# Preparations of *trans*- and *cis*-*μ*-1,2-Peroxodiiron(III) Complexes

**DOI:** 10.3390/molecules29010205

**Published:** 2023-12-29

**Authors:** Yuji Kajita, Masaki Kubo, Hidekazu Arii, Shinya Ishikawa, Yamato Saito, Yuko Wasada-Tsutsui, Yasuhiro Funahashi, Tomohiro Ozawa, Hideki Masuda

**Affiliations:** 1Department of Applied Chemistry, Graduate School of Engineering, Aichi Institute of Technology, 1247 Yachigusa, Yakusa-cho, Toyota 470-0392, Japan; ykaji1974@aitech.ac.jp; 2Department of Frontier Materials, Graduate School of Engineering, Nagoya Institute of Technology, Gokiso, Showa, Nagoya 466-8555, Japan; m.kubo.239@stn.nitech.ac.jp (M.K.); wasada.yuko@nitech.ac.jp (Y.W.-T.); ozawa.tomohiro@nitech.ac.jp (T.O.); 3Department of Education, Graduate School of Education, University of Miyazaki, Gakuenkibanadai-Nishi, Miyazaki 889-2192, Japan; hidekazu.arii@cc.miyazaki-u.ac.jp; 4Department of Chemistry, Graduate School of Science, Osaka University, Machikaneyama, Toyonaka, Osaka 560-0043, Japan; funahashi@chem.sci.osaka-u.ac.jp

**Keywords:** *cis*-*μ*-1,2-peroxodiiron complex, *trans*-*μ*-1,2-peroxodiiron complex, resonance Raman spectrum, X-ray crystal structure, DFT calculation

## Abstract

The iron(II) complex with *cis,cis*-1,3,5-tris(benzylamino)cyclohexane (Bn_3_CY) (**1**) has been synthesized and characterized, which reacted with dioxygen to form the peroxo complex **2** in acetone at −60 °C. On the basis of spectroscopic measurements for **2**, it was confirmed that the peroxo complex **2** has a *trans*-*μ*-1,2 fashion. Additionally, the peroxo complex **2** was reacted with benzoate anion as a bridging agent to give a peroxo complex **3**. The results of resonance Raman and ^1^H-NMR studies supported that the peroxo complex **3** is a *cis*-*μ*-1,2-peroxodiiron(III) complex. These spectral features were interpreted by using DFT calculations.

## 1. Introduction

Diiron-containing proteins are known to play an important role in the respiratory and metabolic systems. Especially, oxidoreductases such as soluble methane monooxygenase (sMMO) and ribonucleotide reductase (RNR), with high-valent dinuclear Fe_2_(*μ*-O)_2_ species as intermediates, are very important from the viewpoint of organic, catalytic and industrial chemistries [1,2,3,4,5,6,7,8,9], because the diiron(III)-peroxo species as precursors for the Fe_2_(*μ*-O)_2_ species are a fundamental key for cleaving the O–O bond of the dioxygen molecule. Therefore, a number of *μ*-peroxodiiron(III) complexes have been synthesized and investigated spectroscopically and structurally by many bioinorganic chemists [5,10,11,12,13,14,15,16,17,18,19,20,21,22,23,24,25,26,27,28,29,30]. In all the crystal structures of *μ*-peroxodiiron(III) complexes reported previously, the torsion angles of Fe–O–O–Fe are in the range of 0° to 53° [12,13,14,15], defined as a *cis*-*μ*-1,2 mode. The torsion angle is restricted to the bridging moiety and supporting ligand such as an alkoxide and a carboxylate, because the removal of the bridging agent induces the preparation of (*μ*-peroxo)diiron(III) species with a larger Fe–O–O–Fe torsion angle. Characterization of various (*μ*-peroxo)diiron(III) species aids the understanding for the mechanism and active intermediates in enzyme reactions such as sMMO and RNR [3,5,10,11,31]. In recent years, the preparation, characterization and reactivity of several peroxo-bridged diiron complexes have received particular attention and researched as precursors to highly active Fe(IV)=O species, which are considered as candidates for active intermediates in sMMO. Goldberg et al. prepared peroxo-bridged diiron(III) species using the siloxide-coordinated ferrous complex Fe^II^(Me_3_TACN)((OSiPh_2_)_2_O) [29] and the thiolate-coordinated ferrous complex Fe^II^(Me_3_TACN)(S_2_SiMe_2_) [30] to study the reactivities of the highly active Fe(IV)=O species. The *μ*-1,2-peroxo-diferric intermediate P of the non-heme diiron enzyme has also attracted considerable attention, as it has been proposed that it can be converted to a high-valence active species by protonation or via a hydroperoxo-diiron(III) intermediate to an activated P′ intermediate. T. Glaser et al. also prepared the *μ*-1,2-peroxo complex {Fe^III^(*μ*-O)(*μ*-1,2-O_2_)Fe^III^} using a dinucleating ligand and studied its reactivity [32]. In the course of oxidation and protonation studies, they succeeded in preparing unique *μ*-1,2-peroxo Fe^IV^Fe^III^ and *μ*-1,2-hydroperoxo-Fe^III^Fe^III^ species. However, in both cases, no distinction was made between *cis* and *trans μ*-peroxo species, much less an assessment of differences in reactivity. At this point, all complexes studied so far have bridging ligands, and there are rare examples of (*μ*-peroxo)diiron(III) species without any bridging agents. It is also quite significant to prepare a *μ*-peroxodiiron(III) complex with other coordination modes such as *trans*-*μ*-1,2 one, because the *trans*-*μ*-1,2-peroxo diiron(III) species has been proposed as one of plausible structures of the reaction intermediate, compound P, in sMMO [3]. So, we considered that the bridging agents of alkoxide and/or calboxylate in the diiron complexes reported hitherto forced the peroxo to behave in a *cis*-*μ*-1,2 fashion, although the carboxylates of glutamic and aspartic acids in sMMO and RNR [3,5,10,11,31] can move flexibly and bind to the irons with longer Fe–O bonds in comparison with those of model complexes [33,34].

In this study, we designed and prepared a tridentate ligand with secondary amino groups, *cis*,*cis*-1,3,5-tris(benzylamino)cyclohexane (Bn_3_CY). Using this ligand, we tried to prepare the *trans*-*μ*-1,2-peroxodiiron(III) species through the dimerization of a mononuclear iron(II) complex. The benzyl group is sterically suitable in the dimerization of the iron(II) complex relative to aliphatic substituents (Figure 1). Moreover, the addition of benzoate is examined to investigate the influence for peroxo species. In the presence of a bridging ligand such as benzoic acid, the iron(II) complex promotes the generation of *cis*-*μ*-1,2-peroxo diiron(III) species by its reaction with dioxygen, an approach studied by M. Suzuki et al. [35] and P. Stavropoulos et al. [36]. On the other hand, in the absence of the bridging ligand, the iron(II) complex with two bulky monodentate ligands such as chloride ion may cause the generation of the *trans*-*μ*-1,2-peroxo diiron(III) complex by the reaction with dioxygen. This *trans*-*μ*-1,2 mode will be changed to *cis*-*μ*-1,2 mode by the addition of benzoic acid.

## 2. Results and Discussion

### 2.1. Synthesis of [Fe^II^(Bn_3_CY)Cl_2_] (**1**)

The starting material, mononuclear iron(II) complex [Fe^II^(Bn_3_CY)Cl_2_] (**1**), was obtained as a white powder by reaction of Bn_3_CY with (*n*-Bu_4_N)_2_[FeCl_4_] in EtOH under Ar. The recrystallization in CH_2_Cl_2_/Et_2_O for one day afforded a colorless single crystal suitable for X-ray analysis. The crystal structure of **1** (Figure 1) revealed that the iron(II) ion is a square pyramid coordinated with three secondary amine nitrogen atoms of Bn_3_CY and two chloride ions. The bond lengths of Fe–N and Fe–Cl are similar to those of mononuclear iron(II) complexes reported hitherto [37,38]. The estimated τ value [39], 0.32, indicates that the geometry around the iron ion is a slightly distorted square pyramidal with a vacant site, which was caused by steric repulsion of two chloride ions.

### 2.2. Formation of the Peroxodiiron Complex **2**

An acetone solution of **1** did not give any band in the near-ultra-violet and visible regions. Exposure of the solution to dioxygen at –60 °C afforded new species (**2**) with two intense bands at 385 nm (sh, *ε* = 2800 M^−1^ cm^−1^ per Fe) and 640 nm (*ε* = 870 M^−1^ cm^−1^ per Fe), as shown in Figure 2, which was EPR silent at 77 K due to a strong antiferromagnetic interaction between diiron(III) centers. The band at 648 nm for **2** is assignable to an LMCT band of peroxo to iron(III), which lies within the range of those (500–700 nm) for *μ*-peroxodiiron(III) complexes reported previously [3]. Another band at 385 nm is comparable to the LMCT band (382 nm (*ε* = 3170 M^−1^ cm^−1^)) of chloride ions to an iron(III) in [Fe^III^(TPA)Cl_2_]ClO_4_ [40], indicating that the chloride ions are maintained for the coordination to iron(III) in **2**. The intensity change in the band at 648 nm indicates that the formation of **2** from **1** at –60 °C obeys a pseudo first-order kinetic law, and the observed rate constant was estimated to be 2.8(3) × 10^−3^ s^−1^ (Appendix A). The kinetic behavior for the dimer-peroxo system was reported by Itoh and Fukuzumi et al., in which the (superoxo)copper(II) species formed by the reaction of a mononuclear copper(I) complex with dioxygen followed by the dimerization of another copper(I) one yields the (*μ*-peroxo)dicopper(II) species. It is concluded that the formation process of the (*μ*-peroxo)dicopper(II) species is a rate-determining step because the formation rate of (*μ*-peroxo)dicopper(II) species is second order to the concentration of the copper(I) complex as a starting material [41]. According to the results, the formation process of a putative (superoxo)iron(III) species in this system is suggested to be the rate-determining step.

### 2.3. Resonance Raman Spectroscopy

The resonance Raman spectrum of **2** in acetone-*d*_6_ gave a characteristic band of O–O stretching vibration at 842/854 cm^−1^ (Figure 3), which shifted to 793/803 cm^−1^ using ^18^O_2_ instead of ^16^O_2_, in which the two bands may be explained in terms of the Fermi doublet or the existence of two diastereomers (Appendix A). The *ν*(O–O) bands appeared in the lowest region among those (888, 900 cm^−1^) of *cis*-*μ*-1,2-peroxodiiron(III) complexes without bridging *μ*-oxo [13,15]. Such a small value in *ν*(O–O) corresponds well to those (832–837 cm^−1^ and 788 cm^−1^ for ^16^O- and ^18^O-labeled samples, respectively) of the dicopper(II) complexes with peroxo in a *trans*-*μ*-1,2 fashion reported previously by Karlin et al. and Suzuki et al. [42,43,44], although they are *μ*-peroxo dicopper(II) complexes. These facts suggest that the peroxide coordinates to the two iron(III) centers in a *trans*-*μ*-1,2 fashion. The oxygenation process is considered as follows (Figure 1): **1** reacts with dioxygen to form a mononuculear iron(III)-superoxo species and then immediately the iron(III)-superoxo complex binds with another **1** to afford **2**. The above kinetic experiment indicates that the first step is the rate-determining step in the formation of **2.**

### 2.4. Conversion of **2** to **3**

In order to confirm whether carboxylate affects the peroxo coordination mode, as described above, a mixture of benzoic acid (150 eq) and triethylamine (150 eq) in acetone was added to the acetone solution of **2** at −60 °C to proceed the complete conversion. The LMCT band of peroxo to iron(III) shifted to a shorter-wavelength region at 595 nm (*ε* = 560 M^−1^ cm^−1^ per Fe) and that of chloride to iron(III) disappeared (Figure 2), indicating that **2** was converted to a new peroxodiiron(III) complex (**3**) and that the chloride ions coordinated to iron(III) atoms were replaced by benzoate. In the resonance Raman spectrum, the O–O stretching vibration observed in **2** at 842/854 cm^−1^ shifted to 865 cm^−1^, which shifted to 818 cm^−1^ when ^18^O_2_ was used instead of ^16^O_2_ (Figure 3). The *ν*(O–O) band is assignable to that of the *cis*-*μ*-1,2-peroxodiiron(III) complex, although it is slightly smaller as compared to those (888, 900 cm^−1^) of *cis*-*μ*-1,2-peroxodiiron(III) complexes reported previously [13,15]. This behavior demonstrates that the peroxide of **2** in a *trans*-*μ*-1,2 fashion was changed to **3** in *cis* by the bridge of exogenous benzoate to the diiron(III) core, indicating that the bridging carboxylate ligand is a factor that compels the diiron complex to *cis*-*μ*-1,2 form. In order to understand the reaction behaviors, we tried ^1^H-NMR and ESI-MS spectral measurements. The ^1^H-NMR study gave interesting spectra, as described below, although we failed the characterizations of **2** and **3** by ESI-MS.

### 2.5. ^1^H-NMR Studies

The ^1^H-NMR spectra of **2** and **3** were measured in acetone-*d*_6_/CH_2_Cl_2_ (24:5 *v*/*v*), and the magnetic moments were calculated from the difference in proton chemical shifts of CH_2_Cl_2_ between in the complex solution and in solvent only by using the Evans method [45]. Both spectra gave extensively shifted and broadened ones typical of a paramagnetic species, as shown in Figure 4 and Figure 5, respectively. These spectral behaviors were also supported from the magnetic moments of *μ*_eff_/*μ*_B_ = 5.4 and 5.3 per one iron atom in species **2** and **3**, respectively. Furthermore, interestingly, species **2** indicated a simple spectrum (seven proton peaks), although the peaks were not assigned, indicating that the species is symmetric. On the other hand, species **3** gave a slightly complicated spectrum (18 proton peaks), suggesting that the species is asymmetric. The respective peak numbers are quite reasonable when we consider the ideal structures of dioxygen diiron complexes with *trans*- and *cis*-conformations. So, we decided that complexes **2** and **3** exist as *trans* and *cis* structures, respectively, in solution.

### 2.6. DFT Calculations

In order to evaluate the observed vibrational modes of species **2** and **3** and to make sure of the reaction mechanism shown in Figure 1, DFT calculations were carried out. We first carried out a preliminary DFT calculation using four small models before carrying out the full geometry optimizations of **2** and **3**. Among them, the monobenzoato *cis*-diiron species was stabilized in the form with peroxide anion in the side-on mode, and the frequency of the O–O stretching modes was quite low, which did not coincide with the experimental results. The formation of the side-on peroxide species in the monobenzoato *cis*-diiron small model may be attributed to the lack of the steric hindrance of the bulky benzyl and benzoate groups. The other three cases gave reasonable results. Additionally, taking the NMR results, where species **2** and **3** are paramagnetic, into consideration, we treated only the complexes with larger coordination numbers of chlorido and benzoato ligands for the calculations of species **2** and **3**, respectively, i.e., the tetrachlorido species for **2** and the dibenzoato species for **3** (Figure 1). The calculations were performed for *trans*- and *cis*-species, **2** and **3**, in the *S* = 0 and *S* = 5 states. 

The structural parameters related to the bridged peroxide and the Raman frequencies and activities of the modes related to O–O stretching for the fully optimized species **2** and **3** in the *S* = 0 and *S* = 5 states are listed in Table 1. The calculations showed that species **2** and **3** in the *S* = 5 state are less stable than those in the *S* = 0 state by 1.3 and 1.7 kcal/mol, respectively (Table 1). On the other hand, the total spins (*S*) estimated for species **2** and **3** in the *S* = 0 state included a large spin contamination, although those in the *S* = 5 state did not, which caused an overestimate in the stability of species in the *S* = 0 state. These results, as expected in the NMR study of species **2** and **3**, suggest that the species in the *S* = 5 state may give better results than those in the *S* = 0. So, we mainly describe and discuss only the structural parameters and Raman frequencies of species **2** and **3** in the *S* = 5 state. The optimized structures of species **2** and **3** are shown in Figure 6. As expected, species **2** and **3** were stabilized in *trans*-O_2_ and *cis*-O_2_ bridged structures, respectively. The Fe–O–O–Fe dihedral angles are 180.0° and 72.1° for **2** and **3**, respectively. The dihedral angle calculated for small models of species **2** (*S* = 5), 129.4°, which is closer to the dihedral angle in gas phase H_2_O_2_ (119.8°) [46], was largely changed from the initial structure, although that of species **3** (*S* = 5) was not significantly changed. It is considered that the steric effect of the benzyl groups of the Bn_3_CY ligand and the phenyl groups of the benzoate stabilized the *trans*-O_2_ structure. Bis-*μ*-benzoate bindings formed in species **3** might be rigid enough to keep the *cis*-O_2_ structure from the steric effect.

The optimized Fe···Fe distances for species **2** and **3**, 4.668 and 3.734 Å, are reasonable when we consider the typical *trans*- and *cis*-O_2_ bridged structures, respectively. In these cases, the O–O bond lengths were estimated to be 1.389 and 1.407 Å for **2** and **3**, respectively. The optimized structure obtained here is also supported by the previously reported structure of the peroxoiron dinuclear complex [35,36].

The Raman frequency calculated for species **2** (*S* = 5) was obtained as two peaks at 917 and 926 cm^−1^, which are very interesting and important. These two signals were also detected in the experimental Raman spectrum (842, 854 cm^−1^, Figure 3), although they were slightly higher. Furthermore, interestingly, their peak heights are similar to the experimental values; Raman activities are estimated to be 1.1 × 10^4^ and 2.1 × 10^4^ for 917 and 926 cm^−1^, respectively. These two peaks consist of the O–O stretching mode and vibration mode of ligands, and the Raman activity increases as the contribution of O–O stretching modes increases. For **3**, the calculated value assigned to O–O stretching mode, 894 cm^−1^, was one signal, which is very similar to the experimental one, 865 cm^−1^.

For species **2**, the O–O stretching mode is coupled with the motion of benzyl groups of the Bn_3_CY ligands (Appendix A). For species **3** in the *S* = 0 state, the O–O stretching mode is coupled with the vibrational modes of the cyclohexyl ring of Bn_3_CY (Appendix A). In contrast, only one O–O stretching mode found in species **3** in the *S* = 5 state, where the O–O bond is longer than that of species **2** (Table 1), results from small vibrational coupling. These results agree with both the experimental Raman spectra and magnetic susceptibility of species **3**.

## 3. Experimental Section

### 3.1. Experimental Procedure

All manipulations were carried out under an atmosphere of purified Ar in a mBRAUN MB 150B-G glovebox or standard Schlenk techniques.

### 3.2. Materials

Reagents and solvents employed were commercially available. All anhydrous solvents were purchased from FUJIFILM Wako Pure Chemical Corporation and were purged with Ar to degas. The ligand *cis*,*cis*-1,3,5-tris(benzylamino)cyclohexane (Bn_3_CY) was prepared following a method from the literature [47].

### 3.3. Preparation of [Fe^II^Cl_2_(Bn_3_CY)] (**1**)

To an EtOH solution (3 mL) of *tetra*-*n*-butylammonium chloride (247 mg, 0.89 mmol) was added a methanol solution of FeCl_2_ (56.3 mg, 0.44 mmol), where insoluble material was removed by filtration. After Bn_3_CY 177 mg (0.44 mmol) was added to the filtrate, the solution stood for a few minutes to obtain a white powder, 199 mg (97%). Calcd for **1**·H_2_O·1.5EtOH (C_30_H_44_Cl_2_FeN_3_O_2.5_): C 58.74, H 7.23, N 6.85. Found: C 58.81, H 7.32, N 6.77.

Leaving the CH_2_Cl_2_/Et_2_O mixed solution of the powder to stand for a few days afforded a colorless single crystal suitable for X-ray analysis.

CCDC reference number CCDC-248487.

### 3.4. Preparations of trans- (2) and cis-μ-1,2-Peroxo Diiron(III) Complexes (3)

The Fe(II) complex **1** was dissolved in acetone to a concentration of 0.4 mM under Ar atmosphere, and then dioxygen was flowed at −60 °C to prepare the peroxo-diiron(III) complex (**2**). The solution color was changed from light yellow to blue green. The UV-vis spectrum of the solution was measured at −60 °C. It was also dissolved in acetone or acetone-*d*_6_ for resonance Raman spectra and further dissolved in acetone-*d*_6_/CH_2_Cl_2_ (24/5 *v*/*v*) solution for ^1^H NMR measurements.

Next, the solution of complex **2** was then prepared in the same way as above, and 10 equivalents of benzoic acid and triethylamine were added to the acetone or acetone-*d*_6_ solution of complex **2** to obtain complex **3**. Various spectroscopic measurements were carried out on the complex solution in the same way as described above.

Unfortunately, complexes **2** and **3** decompose quickly unless the temperature is below −60 °C.

### 3.5. Instrumentation

^1^H-NMR spectral measurements were performed on a BRUKER ADVANCE 600 MHz NMR spectrometer operating at 600.134 MHz (^1^H) in CDCl_3_ at 293 K. Chemical shifts are recorded with respect to internal TMS (δ = 0.00 ppm). Paramagnetic ^1^H-NMR spectra were taken with a BRUKER ADVANCE 400 MHz NMR spectrometer operating at 400.13 MHz (^1^H) in acetone-*d*_6_/CH_2_Cl_2_ (24:1 *v*/*v*) at −60 °C, and magnetic susceptibility was estimated in acetone-*d*_6_/CH_2_Cl_2_ (24:5 *v*/*v*) at −60 °C by using Evans method [45,46,48,49,50]. Electronic absorption spectra were recorded on a JASCO V-570 spectrophotometer (JASCO Corporation, Tokyo, Japan). IR spectra of solid compounds were measured as KBr pellets using a JASCO FT/IR-410 spectrophotometer (JASCO Corporation, Tokyo, Japan). Raman scattering was excited at 600 nm with a He-Cd laser (MKS Instrument Inc., Tokyo, Spectra Physics, Model CD4805R) and detected by using a CCD detector (Teledyne Princeton Instruments, USA, PI-CCD) attached to a single polychromator (Ritsu Oyo Kogaku, Japan, DG-1000). The slit width was set to 200 μm. The laser power used was 30 mW at the sample point. All measurements were carried out with a spinning cell (3000 rpm) at −80 °C. Raman shifts were calibrated with toluene and CH_2_Cl_2_, and the accuracy of the peak positions of the Raman bands was ±1 cm^−1^. Electrospray ionization time-of-flight mass spectra ESI-TOF/MS were obtained with a Micromass LCT spectrometer. X-band EPR spectra were obtained using a JEOL RE-1X spectrometer at 77 K. Elemental analyses were obtained with a Perkin Elmer CHN-900 elemental analyzer (USA).

### 3.6. X-ray Crystallography

A single crystal of **1** suitable for X-ray diffraction analyses was obtained from CH_2_Cl_2_/Et_2_O mixed solution after standing for a few days under an Ar atmosphere. The crystal was mounted on a glass fiber, and diffraction data were collected on a Rigaku/MSC Mercury CCD using graphite monochromated Mo *K*α radiation at 173 K. Crystal data and experimental details are listed in Appendix A.

The structure was solved by a combination of direct method (SIR2004) and Fourier techniques. All non-hydrogen atoms were refined anisotropically. Hydrogen atoms were refined using the riding model. A Sheldrick weighting scheme was employed. Plots of ∑*w* (|*F*_0_| − |*F*_c_|)^2^ versus |*F*_0_|, reflection order in data collection, sin *θ*/λ, and various classes of indices showed no unusual trends. Neutral atomic scattering factors were obtained from Cromer and Waber [51]. Anomalous dispersion terms were included in *F*_calc_ [52], and the values for Δ*ϕ*′ and Δ*ϕ*″ were those of Creagh and McAuley [53]. The values for the mass attenuation coefficients are those of Creagh and Hubbell [54]. All calculations were performed using the crystallographic software package CrystalStructure 3.7.0 [55,56]. A packing diagram of the unit cell and a complete set of crystallographic tables, including positional parameters, anisotropic thermal factors, and bond lengths and angles, are listed in Appendix A.

### 3.7. Computational Details

To assign the observed vibrational modes of species **2** and **3** and to make sure of the reaction mechanism shown in Figure 1, the DFT calculations were carried out using the B3LYP functional [57]. For both cases, the calculations were performed for diiron(III) complexes having the antiferromagnetically and ferromagnetically coupled ferric centers (in the *S* = 0 and *S* = 5 states per one iron site, respectively), although the spin states of iron sites in the solution were estimated to be *μ*_eff_/*μ*_B_ = 5.4 and 5.3 for species **2** and **3**, respectively, on the basis of Evans method using NMR measurement. For the preliminary DFT calculations, we calculated for the following four small model species that the phenyl group of benzoate anion and the benzyl groups of the ligands were replaced with H atoms: dichloride and tetrachloride forms for species **2** and monobenzoate and dibenzoate forms for species **3**. The initial structures of *cis*- and *trans*-species were carried out using the ideal *cis*- and *trans*-conformations, which were geometrically optimized. And, then, whole structures of species **2** and **3** were calculated on the basis of the structures estimated from small models. The following basis sets were employed for the respective atoms; 6-311+G(d) for Fe [58,59], O and Cl atoms [60,61] of an amide group, 6-311G(d) for N atoms [60,62], and 6-31G(d) for C and H atoms [60,63], respectively. The calculated frequencies were scaled by 0.966 [64]. All electronic structure calculations were performed with the Gaussian 09 Rev. E. 01 [65] on the Fujitsu CX400 system at the Nagoya University Information Technology Center.

## 4. Conclusions

We synthesized a mononuclear iron(II) complex **1** with tridentate secondary amine ligand Bn_3_CY. The reaction of **1** with dioxygen at –60 °C in acetone afforded the *trans*-*μ*-1,2-peroxodiiron(III) complex **2**, which was changed to the *cis*-*μ*-1,2 one by the addition of benzoate. These findings suggest that in diiron core-containing proteins, the carboxylate residues must freely control the coordination state of the peroxo intermediate in the enzymatic reaction. ^1^H-NMR measurements of **2** and **3** suggested that these complexes are a paramagnetic species with *S* = 5/2 per one iron and that complexes **2** and **3** are symmetric and asymmetric, respectively, with the findings supporting their conformations. The theoretical explanations for the formations of *cis*- and *trans*-*μ*-1,2-peroxo diiron complexes were elicited using the DFT method.

## Data Availability

The data presented in this study are available on request from the corresponding author.

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
