# Peer review of "Preparations of *trans*- and *cis*-*μ*-1,2-Peroxodiiron(III) Complexes"

_molecules, 2023, doi:10.3390/molecules29010205_

Round 1

Reviewer 1 Report

Comments and Suggestions for Authors

The work describes reaction of O2 with a ferrous complex bearing a tripodal cyclohexyl triamine ligand. The initial reaction results in a peroxide complex that is caharcterized as a trans-peroxide on the basis of its resonance raman, 1H-NMR and DFT calculation. Reaction with benzoates changes the structure of the complex, generating a new peroxide where the bridging mode of the peroxide is proposed to have changed to cis-.

The trends in the energy of the O-O vibrations determined by Raman agrees well with the calculated data, and the symmetry of the complexes derived from 1H-NMR is also consistent with the authors interpretations, despite the proposal has an important degree of speculation. For example, one wonders what happens with the chloride ligands. Is there any evidence that they have detached from the complex?

I would suggest the authors to employ a benzoic acid tagged with a F atom (or a CF3 group) in order to analyze the structure of the complexes by F-NMR. The same strategy can be used by labelling the arene arms of the ligand. This will greatly facilitate the NMR analysis.

In any case, these re quite interesting molecules for teh field of bioinorganic chemistry and publication of the work is therefore recommended.

Author Response

Please look at the attached file (the reply to the Reviewer 1).

Reviewer 2 Report

Comments and Suggestions for Authors

The paper of Hideki Masuda and co-authors is a very interesting fundamental work on synthesis new Iron(II) chloride complex with polydentate N-donor ligand and studying the reactivity towards molecular oxygen. The authors determined the molecular structure of the initial iron complex, and proved the formation of complexes 2 and 3 with peroxo-bridges by indirect methods. If the authors could isolate complexes 2 and 3 in crystal form, that would be great, and the value of the article would be much higher.

 The Cambridge Structural Database contains only 22 deposited structures with the iron-oxygen-oxygen-iron fragment. Synthesis and characterization of such objects is a chalenge task. The authors, in my personal opinion, took the main idea of the study from these previously published articles:

 Structural and Spectroscopic Characterization of (u-Hydroxo or 4-Oxo)(u-peroxo)diiron(III)

Complexes: Models for Peroxo Intermediates of Non-Heme Diiron Proteins 10.1021/ja045594a

Generation of a μ-1,2-hydroperoxo FeIIIFeIII and a μ-1,2-peroxo FeIVFeIII Complex

https://doi.org/10.1038/s41467-022-28894-5

 And Crystal Structure Analysis of a Synthetic Non-Heme DiironO2 Adduct: Insight into the Mechanism of Oxygen Activation https://doi.org/10.1002/anie.199606181

 the authors cited two papers, but not the article in Natural Communications

 Previously obtained data on the structure of molecular iron complexes with peroxo-bridges (https://doi.org/10.1039/C7CC04382A and https://doi.org/10.1021/ic000261+)) can indirectly confirm the optimized structure obtained using DFT calculations by the authors of the current manuscript.

 Based on the analysis of previously published works and the meticulous study of complexes 2 and 3 by the authors of the current manuscript, I have no doubt that the authors observed the formation of iron peroxo compounds.

I have only minor comments on the text

Line 55: “bridging ligand such as benzoic acid” - It is probably better to use the expression “benzoic acid anion”, while carboxylate anions do not always act as bridging ligands; it should be clarified that the idea was taken from previously published works on working with iron carboxylate complexes (here https://doi.org/10.1039/C7CC04382A or https://doi.org/10.1021/ic000261+)

Line 75: “reaction of Bn3CY with (n-Bu4N)2[FeCl4] in EtOH under Ar” - Why was tetrabutylammonium chloride taken? Does the ligand not react with the starting ferric chloride 2+?

Did the authors try to make the same reaction with elemental sulfur?

Line 139: “although we failed the characterizations of 2 and 3 by ESI-MS.” Why? Are the complexes unstable at room temperature? Have authors tried “old school” elemental analysis?

 There is a lack of description of the methodology for the synthesis of hypothetical compounds 2 and 3. This can be described in paragraph 3.1 - general experimental procedures

 ˚ ºC ˚C - very minor note, the authors use different signs for "degree" throughout, there is an officially accepted sign, please use it.

 The presented manuscript is a very interesting work and I certainly recommend accepting it for publication in the Molecules. This work will definitely attract a lot of attention from researchers and I hope increase citations for the Molecules.

Author Response

Attached the another pdf.
